# Attribute Prototype Network for Zero-Shot Learning

**Wenjia Xu**[1,3,4] *    **Yongqin Xian**[1]    **Jiuniu Wang**[3,4]    **Bernt Schiele**[1]    **Zeynep Akata**[1,2]

[1] Max Planck Institute for Informatics, Saarland Informatics Campus, Saarbrücken, Germany
[2] Cluster of Excellence Machine Learning, University of Tübingen, Germany
[3] School of Electronic, University of Chinese Academy of Science, Beijing, China
[4] Aerospace Information Research Institute, Chinese Academy of Sciences, Beijing China

## Abstract

From the beginning of zero-shot learning research, visual attributes have been shown to play an important role. In order to better transfer attribute-based knowledge from known to unknown classes, we argue that an image representation with integrated attribute localization ability would be beneficial for zero-shot learning. To this end, we propose a novel zero-shot representation learning framework that jointly learns discriminative global and local features using only class-level attributes. While a visual-semantic embedding layer learns global features, local features are learned through an attribute prototype network that simultaneously regresses and decorrelates attributes from intermediate features. We show that our locality augmented image representations achieve a new state-of-the-art on three zero-shot learning benchmarks. As an additional benefit, our model points to the visual evidence of the attributes in an image, e.g. for the CUB dataset, confirming the improved attribute localization ability of our image representation.

## 1   Introduction

Visual attributes describe discriminative visual properties of objects shared among different classes. Attributes have shown to be important for zero-shot learning as they allow semantic knowledge transfer from known to unknown classes. Most zero-shot learning (ZSL) methods [30, 6, 1, 50] rely on pretrained image representations and essentially focus on learning a compatibility function between the image representations and attributes. Focusing on image representations that directly allow attribute localization is relatively unexplored. In this work, we refer to the ability of an image representation to localize and associate an image region with a visual attribute as locality. Our goal is to improve the locality of image representations for zero-shot learning.

While modern deep neural networks [13] encode local information and some CNN neurons are linked to object parts [53], the encoded local information is not necessarily best suited for zero-shot learning. There have been attempts to improve locality in ZSL by learning visual attention [24, 58] or attribute classifiers [35]. Although visual attention accurately focuses on some object parts, often the discovered parts and attributes are biased towards training classes due to the learned correlations. For instance, the attributes *yellow crown* and *yellow belly* co-occur frequently (e.g. for Yellow Warbler). Such correlations may be learned as a shortcut to maximize the likelihood of training data and therefore fail to deal with unknown configurations of attributes in novel classes such as *black crown* and *yellow belly* (e.g. for Scott Oriole) as this attribute combination has not been observed before.

To improve locality and mitigate the above weaknesses of image representations, we develop a weakly supervised representation learning framework that localizes and decorrelates visual attributes. More specifically, we learn local features by injecting losses on intermediate layers of CNNs and enforce

these features to encode visual attributes defining visual characteristics of objects. It is worth noting that we use only class-level attributes and semantic relatedness of them as the supervisory signal, in other words, no human annotated association between the local features and visual attributes is given during training. In addition, we propose to alleviate the impact of incidentally correlated attributes by leveraging their semantic relatedness while learning these local features.

To summarize, our work makes the following contributions. (1) We propose an attribute prototype network (APN) to improve the locality of image representations for zero-shot learning. By regressing and decorrelating attributes from intermediate-layer features simultaneously, our APN model learns local features that encode semantic visual attributes. (2) We demonstrate consistent improvement over the state-of-the-art on three challenging benchmark datasets, i.e., CUB, AWA2 and SUN, in both zero-shot and generalized zero-shot learning settings. (3) We show qualitatively that our model is able to accurately localize bird parts by only inspecting the attention maps of attribute prototypes and without using any part annotations during training. Moreover, we show significantly better part detection results than a recent weakly supervised method.

## 2   Related work

**Zero-shot learning.** The aim is to classify the object classes that are not observed during training [20]. The key insight is to transfer knowledge learned from seen classes to unseen classes with class embeddings that capture similarities between them. Many classical approaches [30, 6, 1, 50, 41] learn a compatibility function between image and class embedding spaces. Recent advances in zero-shot learning mainly focus on learning better visual-semantic embeddings [25, 50, 16, 5] or training generative models to synthesize features [43, 44, 57, 56, 18, 31]. Those approaches are limited by their image representations, which are often extracted from ImageNet-pretrained CNNs or finetuned CNNs on the target dataset with a cross-entropy loss.

Despite its importance, image representation learning is relatively underexplored in zero-shot learning. Recently, [48] proposes to weigh different local image regions by learning attentions from class embeddings. [58] extends the attention idea to learn multiple channel-wise part attentions. [35] shows the importance of locality and compositionality of image representations for zero-shot learning. In our work, instead of learning visual attention like [58, 48], we propose to improve the locality of image features by learning a prototype network that is able to localize different attributes in an image.

**Prototype learning.** Unlike softmax-based CNN, prototype networks [46, 39] learn a metric space where the labeling is done by calculating the distance between the test image and prototypes of each class. Prototype learning is considered to be more robust when handling open-set recognition [46, 32], out-of-distribution samples [3] and few-shot learning [33, 11, 28]. Some methods [3, 47, 23] base the network decision on learned prototypes. Instead of building sample-based prototypes, [8] dissects the image and finds several prototypical parts for each object class, then classifies images by combining evidences from prototypes to improve the model interpretability. Similarly, [55] uses the channel grouping model [52] to learn part-based representations and part prototypes.

In contrast, we treat each channel equally, and use spatial features associated with input image patches to learn prototypes for attributes. [8, 58, 55] learn latent attention or prototypes during training and the semantic meaning of the prototypes is inducted by observation in a post-hoc manner, leading to limited attribute or part localization ability e.g., [58] can only localize 2 parts. To address those limitations, our method learns prototypes that represent the attributes/parts where each prototype corresponds to a specific attribute. The attribute prototypes are shared among different classes and encourage knowledge transfer from seen classes to unseen classes, yielding better image representation for zero-shot learning.

**Locality and representation learning.** Local features have been extensively investigated for representation learning [14, 40, 27], and are commonly used in person re-identification [34, 38], image captioning [2, 22] and fine-grained classification [52, 10, 51]. Thanks to its locality-aware architecture, CNNs [13] exploit local information intrinsically. Here we define the local feature as the image feature encoded from a local image region. Our work is related to methods that draw attention over local features [17, 34]. Zheng et.al. [52] generates the attention for discriminative bird parts by clustering spatially-correlated channels. Instead of operating on feature channels, we focus on the spatial configuration of image features and improve the locality of our representation.

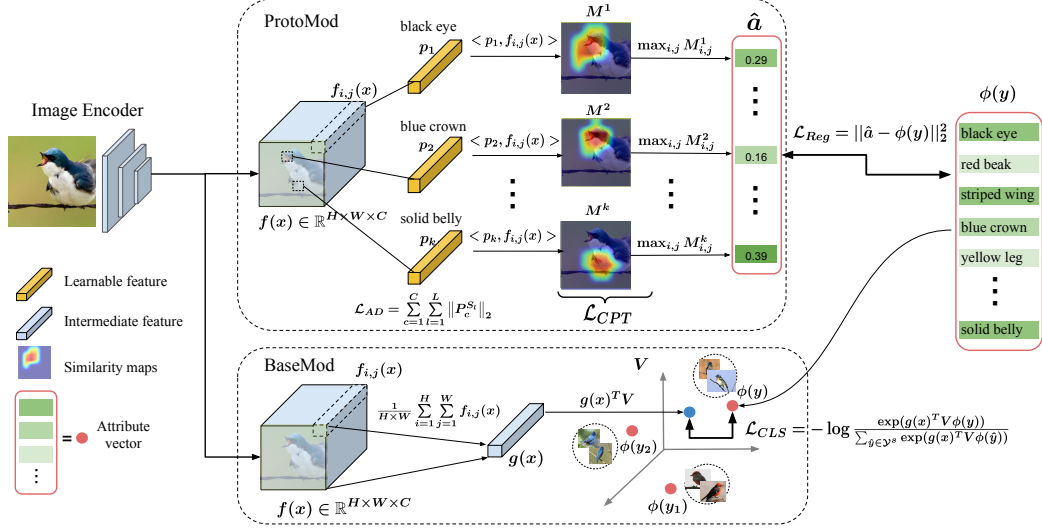

Figure 1: Our attribute prototype network (`APN`) consists of an `Image Encoder` extracting image features $f(x)$, a `BaseMod` performing classification for zero-shot learning, and a `ProtoMod` learning attribute prototypes $p_k$ and localizing them with similarity maps $M^k$. The end-to-end training of `APN` encourages the image feature to contain both global information which is discriminative for classification and local information which is crucial to localize and predict attributes.

## 3 Attribute Prototype Network

Zero-shot learning (ZSL) aims to recognize images of previously unseen classes ($\mathcal{Y}^u$) by transferring learned knowledge from seen classes ($\mathcal{Y}^s$) using their class embeddings, e.g. attributes. The training set consists of labeled images and attributes from seen classes, i.e., $S = \{x, y, \phi(y) | x \in \mathcal{X}, y \in \mathcal{Y}^s\}$. Here, $x$ denotes an image in the RGB image space $\mathcal{X}$, $y$ is its class label and $\phi(y) \in \mathbb{R}^K$ is the class embedding i.e., a class-level attribute vector annotated with $K$ different visual attributes. In addition, class embeddings of unseen classes i.e., $\{\phi(y) | y \in \mathcal{Y}^u\}$, are also known. The goal for ZSL is to predict the label of images from unseen classes, i.e., $\mathcal{X} \rightarrow \mathcal{Y}^u$, while for generalized ZSL (GZSL) [42] the goal is to predict images from both seen and unseen classes, i.e., $\mathcal{X} \rightarrow \mathcal{Y}^u \cup \mathcal{Y}^s$.

In the following, we describe our end-to-end trained attribute prototype network (`APN`) that improves the attribute localization ability of the image representation, i.e. locality, for ZSL. As shown in Figure 1 our framework consists of three modules, the `Image Encoder`, the base module (`BaseMod`) and the prototype module (`ProtoMod`). In the end of the section, we describe how we perform zero-shot learning on top of our image representations and how the locality enables attribute localization.

### 3.1 Base Module (`BaseMod`) for global feature learning

The base module (`BaseMod`) learns discriminative visual features for classification. Given an input image $x$, the `Image Encoder` (a CNN backbone) converts it into a feature representation $f(x) \in \mathbb{R}^{H \times W \times C}$ where $H$, $W$ and $C$ denote the height, width, and channel respectively. `BaseMod` then applies global average pooling over the $H$ and $W$ to learn a global discriminative feature $g(x) \in \mathbb{R}^C$:

$$g(x) = \frac{1}{H \times W} \sum_{i=1}^{H} \sum_{j=1}^{W} f_{i,j}(x), \tag{1}$$

where $f_{i,j}(x) \in \mathbb{R}^C$ is extracted from the feature $f(x)$ at spatial location $(i, j)$ (blue box in Figure 1).

**Visual-semantic embedding layer.** In contrast to standard CNNs with fully connected layers to compute class logits, we further improve the expressiveness of the image representation using attributes. In detail, a linear layer with parameter $V \in \mathbb{R}^{C \times K}$ maps the visual feature $g(x)$ into the class embedding (e.g., attribute) space. The dot product between the projected visual feature

and every class embedding is computed to produce class logits, followed by cross-entropy loss that encourages the image to have the highest compatibility score with its corresponding attribute vector. Given a training image $x$ with a label $y$ and an attribute vector $\phi(y)$, the classification loss $\mathcal{L}_{CLS}$ is:

$$\mathcal{L}_{CLS} = -\log \frac{\exp(g(x)^T V \phi(y))}{\sum_{\hat{y} \in \mathcal{Y}^s} \exp(g(x)^T V \phi(\hat{y}))}. \tag{2}$$

The visual-semantic embedding layer and the CNN backbone are optimized jointly to finetune the image representation guided by the attribute vectors.

## 3.2 Prototype Module (`ProtoMod`) for local feature learning

The global features learned from `BaseMod` may be biased to seen classes because they mainly capture global context, shapes and other discriminative features that may be indicative of training classes. To improve the locality of the image representation, we propose a prototype module (`ProtoMod`) focusing on the local features that are often shared across seen and unseen classes.

**Attribute prototypes.** `ProtoMod` takes as input the feature $f(x) \in \mathbb{R}^{H \times W \times C}$ produced by the `Image Encoder` where the local feature $f_{i,j}(x) \in \mathbb{R}^C$ at spatial location $(i, j)$ encodes information of local image regions. Our main idea is to improve the locality of the image representation by enforcing those local features to encode visual attributes that are critical for ZSL. Specifically, we learn a set of attribute prototypes $P = \left\{ p_k \in \mathbb{R}^C \right\}_{k=1}^K$ to predict attributes from those local features, where $p_k$ denotes the prototype for the $k$-th attribute. As a schematic illustration, $p_1$ and $p_2$ in Figure 1 correspond to the prototypes for *black eye* and *blue crown* respectively. For each attribute (e.g., $k$-th attribute), we produce a similarity map $M^k \in \mathbb{R}^{H \times W}$ where each element is computed by a dot product between the attribute prototype $p_k$ and each local feature i.e., $M_{i,j}^k = \langle p_k, f_{i,j}(x) \rangle$. Afterwards, we predict the $k$-th attribute $\hat{a}_k$ by taking the maximum value in the similarity map $M^k$:

$$\hat{a}_k = \max_{i,j} M_{i,j}^k. \tag{3}$$

This associates each visual attribute with its closest local feature and allows the network to efficiently localize attributes.

**Attribute regression loss.** The class-level attribute vectors supervise the learning of attribute prototypes. We consider the attribute prediction task as a regression problem and minimize the Mean Square Error (MSE) between the ground truth attributes $\phi(y)$ and the predicted attributes $\hat{a}$:

$$\mathcal{L}_{Reg} = ||\hat{a} - \phi(y)||_2^2, \tag{4}$$

where $y$ is the ground truth class. By optimizing the regression loss, we enforce the local features to encode semantic attributes, improving the locality of the image representation.

**Attribute decorrelation loss.** Visual attributes are often correlated with each other as they co-occur frequently, e.g. *blue crown* and *blue back* for Blue Jay birds. Consequently, the network may use those correlations as useful signal and fails to recognize unknown combinations of attributes in novel classes. Therefore, we propose to constrain the attribute prototypes by encouraging feature competition among unrelated attributes and feature sharing among related attributes. To represent the semantic relation of attributes, we divide all $K$ attributes into $L$ disjoint groups, encoded as $L$ sets of attribute indices $S_1, \ldots, S_L$. Two attributes are in the same group if they have some semantic tie, e.g., *blue eye* and *black eye* are in same group as they describe the same body part, while *blue back* belongs to another group. We directly adopt the disjoint attribute groups defined by the datasets [37, 19, 29]. For each attribute group $S_l$, its attribute prototypes $\{p_k | k \in S_l\}$ can be concatenated into a matrix $P^{S_l} \in \mathbb{R}^{C \times |S_l|}$, and $P_c^{S_l}$ is the $c$-th row of $P^{S_l}$. We adopt the attribute decorrelation (AD) loss inspired from [15]:

$$\mathcal{L}_{AD} = \sum_{c=1}^C \sum_{l=1}^L \left\| P_c^{S_l} \right\|_2. \tag{5}$$

This regularizer enforces feature competition across attribute prototypes from different groups and feature sharing across prototypes within the same groups, which helps decorrelate unrelated attributes.

**Similarity map compactness regularizer.** In addition, we would like to constrain the similarity map such that it concentrates on its peak region rather than disperses on other locations.

Therefore, we apply the following compactness regularizer [52] on each similarity map $M^k$,

$$\mathcal{L}_{CPT} = \sum_{k=1}^{K} \sum_{i=1}^{H} \sum_{j=1}^{W} M_{i,j}^k \left[ \left(i - \tilde{i}\right)^2 + \left(j - \tilde{j}\right)^2 \right], \tag{6}$$

where $(\tilde{i}, \tilde{j}) = \arg\max_{i,j} M_{i,j}^k$ denotes the coordinate for the maximum value in $M^k$. This objective enforces the attribute prototype to resemble only a small number of local features, resulting in a compact similarity map.

### 3.3 Joint global and local feature learning

Our full model optimizes the CNN backbone, `BaseMod` and `ProtoMod` simultaneously with the following objective function,

$$\mathcal{L}_{APN} = \mathcal{L}_{CLS} + \lambda_1 \mathcal{L}_{Reg} + \lambda_2 \mathcal{L}_{AD} + \lambda_3 \mathcal{L}_{CPT}. \tag{7}$$

where $\lambda_1$, $\lambda_2$, and $\lambda_3$ are hyper-parameters. The joint training improves the locality of the image representation that is critical for zero-shot generalization as well as the discriminability of the features. In the following, we will explain how we perform zero-shot inference and attribute localization.

**Zero-shot learning.** Once our full model is learned, the visual-semantic embedding layer of the `BaseMod` can be directly used for zero-shot learning inference, which is similar to ALE [1]. For ZSL, given an image $x$, the classifier searches for the class embedding with the highest compatibility via

$$\hat{y} = \arg\max_{\tilde{y} \in \mathcal{Y}^u} g(x)^{\mathrm{T}} V \phi\left(\tilde{y}\right). \tag{8}$$

For generalized zero-shot learning (GZSL), we need to predict both seen and unseen classes. The extreme data imbalance issue will result in predictions biasing towards seen classes [7]. To fix this issue, we apply the Calibrated Stacking (CS) [7] to reduce the seen class scores by a constant factor. Specifically, the GZSL classifier is defined as,

$$\hat{y} = \arg\max_{\tilde{y} \in \mathcal{Y}^u \cup \mathcal{Y}^s} g(x)^{\mathrm{T}} V \phi\left(\tilde{y}\right) - \gamma \mathbb{I}[\tilde{y} \in \mathcal{Y}^s], \tag{9}$$

where $\mathbb{I} = 1$ if $\tilde{y}$ is a seen class and 0 otherwise, $\gamma$ is the calibration factor tuned on a held-out validation set. Our model aims to improve the image representation for novel class generalization and is applicable to other ZSL methods [57, 43, 6], i.e., once learned, our features can be applied to any ZSL model [57, 43, 6]. Therefore, in addition to the above classifiers, we use image features $g(x)$ extracted from the `BaseMod`, and train a state-of-the-art ZSL approach, i.e. ABP [57], on top of our features. We follow the same ZSL and GZSL training and evaluation protocol as in ABP.

**Attribute localization.** As a benefit of the improved local features, our approach is capable of localizing different attributes in the image by inspecting the similarity maps produced by the attribute prototypes. We achieve that by upsampling the similarity map $M^k$ to the size of the input image with bilinear interpolation. The area with the maximum responses then encodes the image region that gets associated with the $k$-th attribute. Figure 1 illustrates the attribute regions of *black eye*, *blue crown* and *solid belly* from the learned similarity maps. It is worth noting that our model only relies on class-level attributes and semantic relatedness of them as the auxiliary information and does not need any annotation of part locations.

## 4 Experiments

We conduct our experiments on three widely used ZSL benchmark datasets. CUB [37] contains $11,788$ images from 200 bird classes with 312 attributes in eight groups, corresponding to body parts defined in [49]. SUN [29] consists of $14,340$ images from 717 scene classes, with 102 attributes divided into four groups. AwA [20] contains 37322 images of 50 animal classes with 85 attributes divided into nine groups [19]. See supplementary Sec.B for more details.

We adopt ResNet101 [13] pretrained on ImageNet [9] as the backbone, and jointly finetune the entire model in an end-to-end fashion to improve the image representation. We use SGD optimizer [4] by setting momentum of $0.9$, and weight decay of $10^{-5}$. The learning rate is initialized as $10^{-3}$ and decreased every ten epochs by a factor of $0.5$. Hyper parameters in our model are obtained by grid search on the validation set [42]. We set $\lambda_1$ as 1, $\lambda_2$ ranges from $0.05$ to $0.1$ for three datasets, and $\lambda_3$ as $0.2$. The factor $\gamma$ for Calibrated Stacking is set to $0.7$ for CUB and SUN, and $0.9$ for AwA.

| Method | ZSL | | | Part localization on CUB | | | | | | |
|---|---|---|---|---|---|---|---|---|---|---|
| | CUB | AWA2 | SUN | Breast | Belly | Back | Head | Wing | Leg | Mean |
| `BaseMod` | 70.0 | 64.9 | 60.0 | 40.3 | 40.0 | 27.2 | 24.2 | 36.0 | 16.5 | 30.7 |
| + `ProtoMod` (only with $\mathcal{L}_{Reg}$) | 71.5 | 66.3 | 60.9 | 41.6 | 43.6 | 25.2 | 38.8 | 31.6 | 30.2 | 35.2 |
| + $\mathcal{L}_{AD}$ | 71.8 | 67.7 | 61.4 | 60.4 | 52.7 | 25.9 | 60.2 | 52.1 | 42.4 | 49.0 |
| + $\mathcal{L}_{CPT}$ (= APN) | **72.0** | **68.4** | **61.6** | **63.1** | **54.6** | **30.5** | **64.1** | **55.9** | **50.5** | **52.8** |

Table 1: Ablation study of ZSL on CUB, AWA2, SUN (left, measuring top-1 accuracy) and part localization on CUB (right, measuring PCP). We train a single `BaseMod` with $\mathcal{L}_{CLS}$ loss as the baseline. Our full model `APN` combines `BaseMod` and `ProtoMod` and is trained with the linear combination of four losses $\mathcal{L}_{CLS}, \mathcal{L}_{Reg}, \mathcal{L}_{AD}, \mathcal{L}_{CPT}$.

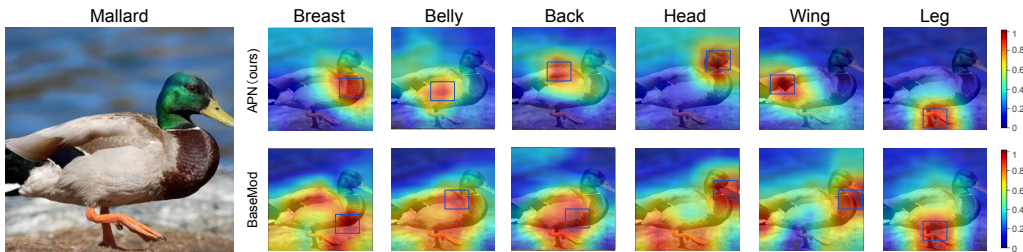

Figure 2: Part localization results: Attention maps for each body part of the bird Mallard generated by our `APN` (first row) and `BaseMod` (second row). Boxes mark out the area with the highest attention. Attention maps are min-max normalized for visualization.

## 4.1 Zero-shot and generalized zero-shot learning results

In the following, we present an ablation study of our framework in the ZSL setting. In addition, we present a comparison with the state-of-the-art in ZSL and GZSL settings.

**Ablation study.** To measure the influence of each model component on the extracted image representation, we design an ablation study where we train a single `BaseMod` with cross-entropy loss as the baseline, and three variants of `APN` by adding the `ProtoMod` and the three loss functions gradually. Our zero-shot learning results on CUB, AWA2 and SUN presented in Table 1 (left) demonstrate that the three additional loss functions improve the ZSL accuracy over `BaseMod` consistently, by 2.0% (CUB), 3.5%(AWA2), and 1.6% (SUN). The main accuracy gain comes from the attribute regression loss and attribute decorrelation loss, which adds locality to the image representation.

**Comparing with the SOTA.** We compare our `APN` with two groups of state-of-the-art models: non-generative models i.e., `SGMA` [58], `AREN` [45], `LFGAA+Hybrid` [26]; and generative models i.e., `LisGAN` [21], `CLSWGAN` [43] and `ABP` [57] on ZSL and GZSL settings. As shown in Table 2, our attribute prototype network (`APN`) is comparable to or better than SOTA non-generative methods in terms of ZSL accuracy. It indicates that our model learns an image representation that generalizes better to the unseen classes. In the generalized ZSL setting that is more challenging, our `APN` achieves impressive gains over state-of-the-art non-generative models for the harmonic mean (H): we achieve 67.2% on CUB and 37.6% on SUN. On AWA2, it obtains 65.5%. This shows that our network is able to balance the performance of seen and unseen classes well, since our attribute prototypes enforce local features to encode visual attributes facilitating a more effective knowledge transfer.

Image features extracted from our model also boosts the performance of generative models that synthesize CNN image features for unseen classes. We choose two recent methods `ABP` [57] and `f-VAEGAN-D2` [44] as our generative models. For a fair comparison, we train `ABP` and `f-VAEGAN-D2` with *finetuned features* [44] extracted from ResNet101, denoted as `ABP`* [57] and `f-VAEGAN-D2`* [44] respectively in Table 2. As for `APN` + `ABP` and `APN` + `f-VAEGAN-D2`, we report the setting where the feature generating models are trained with our *APN feature* $g(x)$. `APN` + `ABP` achieves significant gains over `ABP`*: 2.6% (CUB) and 5.3% (AWA) in ZSL; and 2.7% (CUB), 2.3% (SUN) in GZSL. We also boost the performance of `f-VAEGAN-D2` on three datasets: 0.9% (CUB) and 1.4% (AWA)

| | Method | Zero-Shot Learning | | | Generalized Zero-Shot Learning | | | | | | | | |
| --- | --- | --- | --- | --- | --- | --- | --- | --- | --- | --- | --- | --- | --- |
| | | CUB | AWA2 | SUN | CUB | | | AWA2 | | | SUN | | |
| | | T1 | T1 | T1 | u | s | H | u | s | H | u | s | H |
| §| SGMA [58] | 71.0 | **68.8** | − | 36.7 | 71.3 | 48.5 | 37.6 | 87.1 | 52.5 | − | − | − |
| | AREN [45] | 71.8 | 67.9 | 60.6 | 38.9 | 78.7 | 52.1 | 15.6 | 92.9 | 26.7 | 19.0 | 38.8 | 25.5 |
| | LFGAA+Hybrid [26] | 67.6 | 68.1 | 61.5 | 36.2 | 80.9 | 50.0 | 27.0 | 93.4 | 41.9 | 18.5 | 40.4 | 25.3 |
| | APN (Ours) | **72.0** | 68.4 | **61.6** | 65.3 | 69.3 | **67.2** | 56.5 | 78.0 | **65.5** | 41.9 | 34.0 | **37.6** |
| †| GAZSL [56] | 55.8 | 68.2 | 61.3 | 23.9 | 60.6 | 34.3 | 19.2 | 86.5 | 31.4 | 21.7 | 34.5 | 26.7 |
| | LisGAN [21] | 58.8 | 70.6 | 61.7 | 46.5 | 57.9 | 51.6 | 52.6 | 76.3 | 62.3 | 42.9 | 37.8 | 40.2 |
| | CLSWGAN [43] | 57.3 | 68.2 | 60.8 | 43.7 | 57.7 | 49.7 | 57.9 | 61.4 | 59.6 | 42.6 | 36.6 | 39.4 |
| | ABP* [57] | 70.7 | 68.5 | 62.6 | 61.6 | 73.0 | 66.8 | 53.7 | 72.1 | 61.6 | 43.3 | 39.3 | 41.2 |
| | APN+ABP (Ours) | 73.3 | **_73.8_** | 63.1 | 65.8 | 74.0 | 69.5 | 57.1 | 72.4 | 63.9 | 46.2 | 37.4 | 41.4 |
| | f-VAEGAN-D2* [44] | 72.9 | 70.3 | 65.6 | 63.2 | 75.6 | 68.9 | 57.1 | 76.1 | 65.2 | 50.1 | 37.8 | 43.1 |
| | APN+f-VAEGAN-D2 (Ours) | **_73.8_** | 71.7 | **_65.7_** | 65.7 | 74.9 | **70.0** | 62.2 | 69.5 | **65.6** | 49.4 | 39.2 | **43.7** |

Table 2: Comparing our APN model with the state-of-the-art on CUB, AWA2 and SUN. † and § indicate generative and non-generative representation learning methods respectively. Our model APN uses Calibrated Stacking [7] for GZSL. ABP* and f-VAEGAN-D2* use *finetuned features* extracted from ResNet101. APN+ABP and APN+f-VAEGAN-D2 respectively denote ABP [57] and f-VAEGAN-D2 [44] using our APN features. We measure top-1 accuracy (**T1**) in ZSL, top-1 accuracy on seen/unseen (**s/u**) classes and their harmonic mean (**H**) in GZSL.

in ZSL; and $1.1\%$ (CUB), $0.6\%$ (SUN) in GZSL. It indicates that our *APN features* are better than *finetuned features* over all the datasets. Our features increase the accuracy by a large margin compared to other generative models. For instance, APN + ABP outperforms LisGAN by $14.5\%$ on CUB, $2.2\%$ on AWA2 and $1.4\%$ on SUN in ZSL. These results demonstrate that our learned locality enforced image representation makes better knowledge transfer from seen to unseen classes, as the attribute decorrelation loss alleviates the issue of biasing the label prediction towards seen classes.

## 4.2 Evaluating part and attribute localization

First, we evaluate the part localization capability of our method quantitatively both as an ablation study and in comparison with other methods using the part annotation provided for the CUB dataset. Second, we provide qualitative results of our method for part and attribute localization.

**Ablation study.** Our ablation study evaluates the effectiveness of our APN framework in terms of the influence of the attribute regression loss $\mathcal{L}_{Reg}$, attribute decorrelation loss $\mathcal{L}_{AD}$, and the similarity compactness loss $\mathcal{L}_{CPT}$. Following SPDA-CNN [49], we report the part localization accuracy by calculating the Percentage of Correctly Localized Parts (PCP). If the predicted bounding box for a part overlaps sufficiently with the ground truth bounding box, the detection is considered to be correct (more details in supplementary Sec.C).

Our results are shown in Table 1 (right). When trained with the joint losses, APN significantly improves the accuracy of *breast*, *head*, *wing* and *leg* by $22.8\%$, $39.9\%$, $19.9\%$, and $34.0\%$ respectively, while the accuracy of *belly* and *back* are improved less. This observation agrees with the qualitative results in Figure 2 that BaseMod tends to focus on the center body of the bird, while APN results in more accurate and concentrated attention maps. Moreover, $\mathcal{L}_{AD}$ boosts the localization accuracy, which highlights the importance of encouraging in-group similarity and between-group diversity when learning attribute prototypes.

**Comparing with SOTA.** We report PCP in Table 3. As the baseline, we train a single BaseMod with cross-entropy loss $\mathcal{L}_{CLS}$, and use gradient-based visual explanation method CAM [54] to investigate the image area BaseMod used to predict each attribute (see supplementary Sec.C for implementation details). On average, our APN improves the PCP over BaseMod by $22.1\%$ ($52.8\%$ vs $30.7\%$). The majority of the improvements come from the better leg and head localization.

Although there is still a gap with SPDA-CNN ($52.8\%$ v.s. $73.6\%$), the results are encouraging since we do not need to leverage part annotation during training. In the last two rows, we compare with the weakly supervised SGMA [58] model which learns part attention by clustering feature channels. Under the same bounding box size, we significantly improve the localization accuracy over SGMA ($78.9\%$

| Method | Parts Annotation | BB size | Breast | Belly | Back | Head | Wing | Leg | Mean |
|---|---|---|---|---|---|---|---|---|---|
| SPDA-CNN [49] | ✓ | 1/4 | 67.5 | 63.2 | 75.9 | 90.9 | 64.8 | 79.7 | 73.6 |
| Selective search [36] | | | 51.8 | 51.0 | 56.1 | 90.8 | 62.1 | 66.3 | 63.0 |
| BaseMod (uses $\mathcal{L}_{CLS}$) | ✗ | 1/4 | 40.3 | 40.0 | 27.2 | 24.2 | 36.0 | 16.5 | 30.7 |
| APN (Ours) | | | 63.1 | 54.6 | 30.5 | 64.1 | 55.9 | 50.5 | 52.8 |
| SGMA [58] | ✗ | $1/\sqrt{2}$ | — | — | — | 74.9 | — | 48.1 | 61.5 |
| APN (Ours) | | | 79.4 | 91.9 | 94.7 | 89.7 | 87.2 | 68.0 | 84.7 |

Table 3: Comparing our APN, detection models trained with part annotations (top two rows), and a ZSL model SGMA. BaseMod is trained with $\mathcal{L}_{CLS}$. For BB (bounding box) size, $1/4$ means each part bounding box has the size $\frac{1}{4}W_b * \frac{1}{4}H_b$, where $W_b$ and $H_b$ are the width and height of the bird. For a fair comparison, we use the same BB size as SGMA in last two rows.

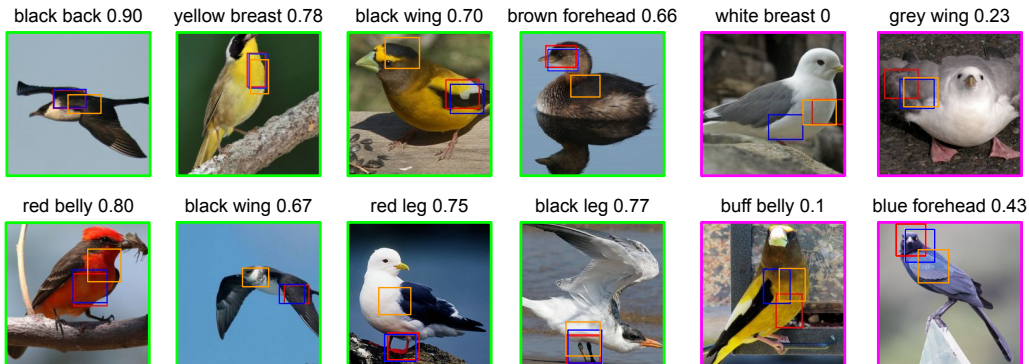

Figure 3: Attribute localization results. Red, blue, orange bounding boxes in the image represent the ground truth part bounding box, the results from our model, and BaseMod+CAM respectively. The number following attribute name is the IoU between ours and the ground truth. Green (purple) box outside the image indicates a correct (incorrect) localization by our model.

v.s. $61.5\%$ on average). Since the feature channels encode more pattern information than local information [12, 53], enforcing locality over spatial dimension is more accurate than over channel.

**Qualitative results.** We first investigate the difference between our APN and the baseline BaseMod for localizing different body parts. In Figure 2, for each part of the bird Mallard, we display one attribute similarity map generated by APN and BaseMod. BaseMod tends to generate disperse attention maps covering the whole bird, as it utilizes more global information, e.g., correlated bird parts and context, to predict attributes. On the other hand, the similarity maps of our APN are more concentrated and diverse and therefore they localize different bird body parts more accurately.

In addition, unlike other models [49, 36, 58] that can only localize body parts, our APN model can provide attribute-level localization as shown in Figure 3. Compared with BaseMod+CAM, our approach produces more accurate bounding boxes that localize the predicted attributes, demonstrating the effectiveness of our attribute prototype network. For example, while BaseMod+CAM wrongly learns the *black wing* from the image region of the head (row 1, column 3 of Figure 3), our model precisely localizes the *black wing* at the correct region of the wings. These results are interesting because our model is trained on only class-level attributes without accessing any bounding box annotation.

As a side benefit, the attribute localization ability introduces a certain level of interpretability that supports the zero-shot inference with attribute-level visual evidences. The last two columns show some failure examples where our predicted bounding boxes achieve low IoUs. We observe that our predicted bounding boxes for the *grey wing* and *blue forehead* are not completely wrong while they are considered as failure cases by the evaluation protocol. Besides, although our attribute decorrelation loss in Equation 5 alleviates the correlation issue to some extent (as shown in the previous results in Table 1 and Figure 2), we observe that our APN seems to still conflate the *white belly* and *white breast* in some cases, indicating the attribute correlation issue as a challenging problem for future research.

| | Attribute Prediction | | Part Localization Accuracy | | | | | | |
|---|---|---|---|---|---|---|---|---|---|
| Attribute type | Seen | Unseen | Breast | Belly | Back | Head | Wing | Leg | Mean |
| APN + Binary | 86.8 | 86.7 | 61.8 | 54.5 | 30.0 | 64.2 | 54.1 | 47.7 | 52.1 |
| APN + Continuous | 86.4 | 85.7 | 63.1 | 54.6 | 30.5 | 64.1 | 55.9 | 50.5 | 52.8 |

Table 4: Attribute prediction accuracy and part localization accuracy of APN model trained with binary attributes ("APN+Binary") or with continuous attributes ("APN+Continuous") on CUB dataset.

**Binary v.s. continuous attributes** The class-level continuous attributes are usually obtained by averaging the image-level binary attributes for each class, which are expensive to collect. In this section, we test if our model can work with class-level binary attributes, which are easier to annotate. Specifically, we retrain our attribute prototype network with class-level binary attributes which are obtained by thresholding the continuous ones. We then evaluate its performance on the part localization and the image-level attribute prediction tasks on the test set of CUB dataset. Our APN model can make an attribute prediction by thresholding the predicted attribute $\hat{a}$ at $0.5$. As shown in Table 4, our method works equally well for both binary and continuous attributes in terms of attribute prediction accuracy ($86.4\%$ with continuous attributes vs $86.8\%$ with binary attributes on seen classes) and part localization accuracy ($52.8\%$ with continuous attributes vs $52.1\%$ with binary attributes). It indicates that our model does not rely on the expensive continuous attribute annotation and generalizes well to the binary attributes that are easier to collect.

## 5  Conclusion

In this work, we develop a zero-shot representation learning framework, i.e. attribute prototype network (APN), to jointly learn global and local features. By regressing attributes with local features and decorrelating prototypes with regularization, our model improves the locality of image representations. We demonstrate consistent improvement over the state-of-the-art on three ZSL benchmarks, and further show that, when used in conjunction with feature generating models, our representations improve over finetuned ResNet representations. We qualitatively verify that our network is able to accurately localize attributes in images, and the part localization accuracy significantly outperforms a weakly supervised localization model designed for zero-shot learning.

## Broader Impact

Computers have become much smarter over the past decades, but they cannot distinguish between two objects without a properly annotated training dataset. Since humans can learn general representations that generalize well across many classes, they require very little or even no training data to recognize novel classes. Zero-shot learning aims to mimic this ability to recognize objects using only some class level descriptions (e.g., identify a bird according to the color and pattern of bird body parts), and takes a first step to build a machine that has a similar decision process as humans. Therefore, ZSL techniques would benefit those who do not have access to large-scale annotated datasets, e.g. a wildlife biologist who wants to build an automatic classification system for rare animals.

Our work introduces an attribute prototype network, which is good at predicting attributes with local features. Specifically, we deal with the attribute correlation problem, where the network cannot tell attributes apart because they co-occur very often, eg, the *yellow forehead* and *yellow crown* of birds. By decorrelating attribute prototypes, we use the local information for predicting attributes, rather than using the correlated contexts, which is an important direction that we hope there will be a greater focus by the community. Additionally, we strengthen the interpretability of the inference process, by highlighting the attributes our model has learnt in the image. Interpretability is important for helping users to understand the learning process and check model errors.

Broadly speaking, there are two shortcuts of zero-shot learning. Firstly, the prediction accuracy is still lower than models trained with both seen and unseen classes. Thus ZSL is not applicable to those settings that require high accuracy and confidence, e.g., self-driving cars. Secondly, the model's generalization ability depends to a great extent on the quality of side information that describes the similarity between seen and unseen classes. Thus biased side information might harm the generalization ability of ZSL models.

## Acknowledgments and Disclosure of Funding

This work has been partially funded by the ERC under Horizon 2020 program 853489 - DEXIM and by the DFG under Germany's Excellence Strategy – EXC number 2064/1 – Project number 390727645. We thank Stephan Alaniz, Max Maria Losch, and Ferjad Naeem for proofreading the paper.

## Footnotes

*xuwenjia16@mails.ucas.ac.cn

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
