[Supplementary Material]

# Attribute Prototype Network for Zero-Shot Learning
# Supplementary material

The supplementary material is organized as follows. In Section A, we present more qualitative results from AWA2, SUN, and CUB datasets. Then, we introduce the attribute group definition in Section B. Finally, we describe the part localization evaluation for CUB dataset in Section C, and apply our feature to the SOTA feature generating methods in Section D.

## A    Qualitative results

As illustrated in Section 3 of the main paper, our approach is able to localize different attributes in the image by inspecting the similarity maps produced by the attribute prototypes. The area with the maximum responses encodes the image region that gets associated with the corresponding attribute. It is worth noting that our model only relies on class-level attributes and semantic relatedness of them as the auxiliary information and does not need any annotation of part locations. Given a certain attribute, we retrieve the top scoring images and show the attribute similarity maps for those images. Specifically, we compute

$$\arg\max_{x} \hat{a}_k(x), \tag{1}$$

where $x$ denotes an image, and $\hat{a}_k(x)$ is the predicted score for the $k$-th attribute of image $x$.

### A.1    AWA2 dataset

In this section, we investigate if our attribute prototype network can localize visual attributes on AWA2 dataset. As shown in Figure A.1, we observe that our network produces precise similarity maps for visual attributes that describe *color*, *texture*, *body parts*, etc. Another interesting observation is that our model is able to localize visual attributes with diverse appearances. For instance, we can locate the white and black *stripe* of zebra, and the yellow and black *stripe* of tiger (row 4). Our model can also locate the *long leg* of giraffe, elephant, and horse (row 8) with different texture and shape. The similarity maps for *furry* can precisely localize the image regions of the ox (row 2, column 3). On the other hand, there are some failure cases which are marked in purple. While on AWA2 dataset we are interested in visual attributes of animals, our model in some cases highlights the attributes of the background e.g., identifying the grid on the rat cage, and the ripples on the water as stripes (row 4, column 5 and 6). This can be explained by the fact that our model only relies on weak supervision i.e., class-level attributes and their semantic relatedness. Overall, those results support our observations in the main paper and indicate that our model is able to perform attribute localization in a weakly supervised manner.

### A.2    SUN dataset

In this section, we discuss the attribute localization results on SUN dataset. Figure A.2 shows the localization results for visual attributes *cloth*, *cloud*, *tree*, and *fencing*, etc. We observe that our model can accurately locate the *cloth* in closet and cloth store, and the cloth on human body (row 2). Further more, we can discriminate between different attributes with similar color or texture, e.g., correctly locating *snow* and *cloud* in one image (row 4, column 3,4), and *tree* and *grass* in one image (row 6, column 4,5). Although the appearance of one visual attribute may vary significantly, we can still locate such attributes correctly, e.g., the *fencing* with different colors, location and shape (row 8, column 1,2,3). There are also some failure cases, e.g., recognizing the *tent* on camp site as *cloth* as they share similar texture, and identifying the flying white *plane* as *cloud*.

Figure A.1: Top scoring images and their attribute similarity maps on AWA2 dataset. We apply min-max normalization on the similarity maps for visualization. We cover the upsampled similarity map on the original image, to show the corresponding location of the highlighted area. The caption above each image indicates the attribute name and (image category). The purple box outside the image indicates an incorrect localization.

Figure A.2: Attribute similarity maps for SUN dataset. Attention maps are min-max normalized for visualization. We cover the upsampled attention map on the original image, to show the corresponding location of the highlighted area. The caption above each image indicates attribute name (image category).

## A.3 CUB dataset

In Figure A.3, we display the attribute similarity maps for Figure 3 in the paper. Our model can accurately localize the attributes of birds, e.g., *black back*, *yellow breast*, and *red belly*, etc. We show more qualitative results from CUB dataset in Figure A.4.

Figure A.3: Attribute similarity maps for CUB dataset. Red and blue bounding boxes on the original image represent the ground truth part bounding box and the results from our model. We cover the upsampled attention map on the original image, to show the corresponding location of the highlighted area. The number following attribute name in the caption is the IoU between ours and the ground truth. Purple box outside the image indicates an incorrect localization.

Figure A.4: More attribute similarity maps for CUB dataset.

| Group | Attributes | Group | Attributes | Group | Attributes |
|---|---|---|---|---|---|
| Belly | Belly Pattern | Head | Bill Shape | Tail | Upper Tail Color |
| | Belly Color | | Bill Color | | Udder Tail Color |
| Breast | Breast Pattern | | Bill Length | | Tail Pattern |
| | Breast Color | | Forehead Color | | Tail Shape |
| | Throat Color | | Nape Color | Others | Bird Size |
| back | Back Color | | Eye Color | | Bird Shape |
| | Back Pattern | | Head pattern | | Primary Color |
| Wing | Wing Color | | Crown Color | | Under Parts Color |
| | Wing Pattern | Leg | Leg Color | | Upper Parts Color |
| | Wing Shape | | | | |

Table B.1: The attribute group definition for CUB dataset. We divided attributes into eight groups, corresponding to seven body parts and an "others" group.

## B   Attribute group definition

We introduce the attribute group definition for three datasets in this section. Following the part definition in SPDA-CNN [9], we define seven parts for all the birds in CUB dataset [6], i.e., *belly*, *breast*, *back*, *wing*, *head*, *leg*, *tail*. As shown in Table B.1, we divide part related attributes into the seven part groups. Other attributes are grouped as "others".

For AWA [3] dataset, we follow [2] to divide 85 attributes into nine groups, describing various properties of animals (see Table B.3), i.e., *color*, *texture*, *shape*, *body parts*, *behaviour*, *nutrition*, *activity*, *habitat* and *character*.

We follow the definition of SUN [5] dataset to divide 102 attributes into four groups (see Table B.2), describing the *functions*, *materials*, *surface properties* and *spatial envelope* of scene images.

## C   Bird Body part localization

### C.1   Part localization accuracy

As illustrated in Section 4.2, our approach is capable of localizing different attributes in the image by inspecting the similarity maps produced by the attribute prototypes. In this section, we explain in detail how we extend the attribute localization to body part localization and evaluate the part localization performance with the part annotation provided in the CUB dataset.

**Part localization.**   As shown in Table C.1, for each body part, there are several attribute subgroups encoded as $M$ sets of attribute indices $G_1, \ldots, G_M$. For instance, *breast* is related to two subgroups, *breast color* and *breast pattern*, while *breast color* subgroup consists of 15 color attributes such as *black breast* and *yellow breast*, etc. Over one attribute subgroup, given the predicted value for each attribute $\{\hat{a}_k | k \in G_m\}$, we evaluate the similarity map for the attribute with highest prediction

$$\underset{k \in G_m}{\arg \max} \ \hat{a}_k. \tag{2}$$

Thus, for each part, we average the part localization accuracy of $M$ similarity maps. The subgroups we evaluated for each body part is shown in Table C.1.

**Evaluation protocol.**   Given the similarity maps related to six body parts, we calculate the Percentage of Correctly Localized Parts (PCP) following SPDA-CNN [9]. The ground truth bounding box $B_g$ for each part is centered by the point annotation and is of size $\frac{1}{4}W_b * \frac{1}{4}H_b$, where $W_b$ and $H_b$ indicate the width and height of the bird bounding box. To generate the predicted bounding box $B_p$ with the same size, we sum the attention values in all possible bounding boxes on similarity map and pick the one with the highest summation of attention value as $B_p$. The part detection of

| Group | Attributes | | | |
|---|---|---|---|---|
| Functions | biking | driving | sailing_boating | transporting |
| | research | diving | swimming | vacationing_touring |
| | gaming | spectating | farming | constructing_building |
| | sunbathing | bathing | hiking | medical_activity |
| | eating | cleaning | socializing | congregating |
| | shopping | climbing | waiting_queuing | using_tools |
| | camping | reading | studying_learning | teaching_training |
| | working | competing | sports | playing |
| | digging | exercise | praying | conducting_business |
| Materials | trees | concrete | running_water | dirt_soil |
| | leaves | flowers | asphalt | pavement |
| | sand | cloth | rubber_plastic | rock_stone |
| | smoke | fire | sterile | scary |
| | grass | vegetation | shrubbery | foliage |
| | metal | paper | wood | vinyl_linoleum |
| | snow | ice | still_water | clouds |
| | fencing | railing | wire | railroad |
| | shingles | carpet | brick | tiles |
| | marble | glass | waves_surf | ocean |
| Surface properties | moist_damp | dry | electric_lighting | aged_worn |
| | glossy | matte | natural_light | direct_sun_sunny |
| | dirty | rusty | | |
| Spatial envelope | warm | cold | open_area | vertical_components |
| | natural | stressful | enclosed_area | far-away_horizon |
| | no_horizon | rugged_scene | cluttered_space | horizontal_components |
| | soothing | man-made | symmetrical | semi-enclosed_area |

Table B.2: Attribute group definition for SUN dataset. The attributes are divided into four groups.

an attribute similarity map is considered to be correct if the predicted bounding box for that part overlaps sufficiently with the ground truth bounding box. We generate two ground truth bounding boxes for wings and legs respectively since birds have two wings and legs. The predicted bounding box of wings/legs will be identified as correct is $B_p$ overlaps sufficiently with any one of the ground truth bounding boxes.

## C.2 BaseMod + CAM

As illustrated in Section 4.2, we train a single `BaseMod` with cross-entropy loss $\mathcal{L}_{CLS}$ as the baseline. In this section, we explain how to apply gradient-based visual explanation method Class Activation Map (`CAM`) [10] to investigate the image region used by the `BaseMod` when predicting each attribute.

As illustrated in Section 3.1, given an input image $x$, the `Image Encoder` converts it into an activation map $f(x) \in \mathbb{R}^{H \times W \times C}$ where $H$, $W$ and $C$ denote the height, width, and channel respectively. We denote the activation map of channel $c$ as $f_c(x) \in \mathbb{R}^{H \times W}$. `BaseMod` then applies global average pooling to $f(x)$ and get a global feature $g(x) \in \mathbb{R}^C$, and learns a visual-semantic embedding layer $V \in \mathbb{R}^{C \times K}$ to map the visual feature $g(x)$ into the attribute space. Thus the predicted value for $k$-th attribute is

$$\tilde{a}_k = \sum_{c=1}^{C} v_c^k g_c(x), \tag{3}$$

where $v_c^k$ is a scalar weight representing the contribution of channel $c$ in predicting the $k$-th attribute and $g_c(x)$ is the $c$-th element of vector $g(x)$. Following CAM [10], the attention map for the $k$-th, attribute is calculated as the weighted sum of channel-wise activation maps,

$$\tilde{M}_k = \sum_{c=1}^{C} v_c^k f_c(x). \tag{4}$$

| Group | Attributes | | | | | |
|---|---|---|---|---|---|---|
| Texture | patches | spots | stripes | furry | hairless | toughskin |
| Shape | big | small | bulbous | lean | | |
| Behaviour | active | inactive | nocturnal | hibernate | agility | |
| Nurition | fish insects | meat forager | plankton grazer | vegetation hunter | scavenger stalker | skimmer newworld |
| Activity | flys walks | hops fast | swims slow | tunnels strong | weak | muscle |
| Character | smelly group | fierce solitary | timid | smart | nestspot | domestic |
| Color | black gray | white orange | blue | brown | red | yellow |
| Body parts | flippers paws chewteeth | hands longleg meatteeth | hooves horns buckteeth | pads bipedal strainteeth | longneck claws quadrapedal | tail tusks |
| Habitat | oldworld bush cave | arctic plains ocean | coastal forest ground | desert fields | water jungle | tree mountains |

Table B.3: Attribute group definition for AWA2 dataset. The attributes are divided into nine groups.

| Body part | Attributes | Body part | Attributes | Body part | Attributes |
|---|---|---|---|---|---|
| Breast | Breast Pattern | | Crown Color | | Wing Pattern |
| | Breast Color | | Eye Color | Wing | Wing Color |
| Back | Back Color | Head | Nape Color | | Wing Shape |
| | Back Pattern | | Forehead Color | Belly | Belly Pattern |
| Leg | Leg Color | | Head Pattern | | Belly Color |

Table C.1: Attributes evaluated for part localization. In total, we evaluate the part localization for six parts.

By upsampling the class activation map $\tilde{M}_k$ to the size of the input image with bilinear interpolation, we can identify the image regions that are most relevant to the attribute $k$.

| Model | Feature | Zero-Shot Learning | | | Generalized Zero-Shot Learning | | | | | | | | |
| | | CUB | AWA2 | SUN | CUB | | | AWA2 | | | SUN | | |
| | | T1 | T1 | T1 | u | s | H | u | s | H | u | s | H |
|---|---|---|---|---|---|---|---|---|---|---|---|---|---|
| CLSWGAN [7] | ResNet101 | 57.3 | 68.2 | 60.8 | 43.7 | 57.7 | 49.7 | 57.9 | 61.4 | 59.6 | 42.6 | 36.6 | 39.4 |
| | APN (Ours) | 71.5 | 68.9 | 62.8 | 61.9 | 74.0 | 67.4 | 60.0 | 65.7 | 63.0 | 44.2 | 38.7 | 41.8 |
| CVC [4] | ResNet101 | 54.4 | 71.1 | 62.6 | 47.4 | 47.6 | 47.5 | 56.4 | 81.4 | 66.7 | 36.3 | 42.8 | 39.3 |
| | APN (Ours) | 71.0 | 71.2 | 60.6 | 0.62 | 74.5 | 67.7 | 63.2 | 81.0 | **71.0** | 37.9 | 45.2 | 41.2 |
| GDAN [1] | ResNet101 | – | – | – | 39.3 | 66.7 | 49.5 | 32.1 | 67.5 | 43.5 | 38.1 | 89.9 | 53.4 |
| | APN (Ours) | – | – | – | 67.9 | 66.7 | 67.3 | 35.5 | 67.5 | 46.5 | 41.4 | 89.9 | **56.7** |
| ABP [11] | ResNet101* | 70.7 | 68.5 | 62.6 | 61.6 | 73.0 | 66.8 | 53.7 | 72.1 | 61.6 | 43.3 | 39.3 | 41.2 |
| | APN (Ours) | 73.3 | **73.8** | 63.1 | 65.8 | 74.0 | 69.5 | 57.1 | 72.4 | 63.9 | 46.2 | 37.4 | 41.4 |
| f-VAEGAN-D2* [8] | ResNet101* | 72.9 | 70.3 | 65.6 | 63.2 | 75.6 | 68.9 | 57.1 | 76.1 | 65.2 | 50.1 | 37.8 | 43.1 |
| | APN (Ours) | **73.8** | 71.7 | **65.7** | 65.7 | 74.9 | **70.0** | 62.2 | 69.5 | 65.6 | 49.4 | 39.2 | 43.7 |

Table D.1: Applying our APN feature on the state-of-the-art feature generating models. "ResNet101" represents the feature extracted from ResNet101 pretrained on ImageNet. "ResNet101*" represents the feature extracted from ResNet101 finetuned on CUB, AWA2 or SUN datasets. APN denotes the feature extracted from our APN network. We measure top-1 accuracy (**T1**) in ZSL, top-1 accuracy on seen/unseen (**s/u**) classes and their harmonic mean (**H**) in GZSL.

# D   Apply to the SOTA

Image features extracted by our model can boost the performance of feature generating methods. In this section, we compare our feature extracted by APN network with features extracted by ResNet101. As shown in Table D.1, our APN feature can improve the zero-shot learning accuracy of CVC [4] by a large margin: $16.6\%$ (CUB) on ZSL; $20.2\%$ (CUB) and $4.3\%$ (AWA2) on GZSL. The improvement on GDAN [1] is also significant: $17.8\%$ (CUB), $3.0\%$ (AWA2) and $3.3\%$ (SUN) on GZSL. For fair comparison, we train ABP [11] and f-VAEGAN-D2 [8] with feature [8] extracted by ResNet101 finetuned on CUB, AWA2 or SUN dataset. Our APN feature is better than finetuned feature over all the datasets. ABP + APN achieves significant gains over ABP + ResNet*: $2.6\%$ (CUB) and $5.3\%$ (AWA) on ZSL; and $2.7\%$ (CUB), $2.3\%$ (SUN) on generalized ZSL. We also boost the performance of f-VAEGAN-D2 on three datasets: $0.9\%$ (CUB) and $1.4\%$ (AWA) on ZSL; and $1.1\%$ (CUB), $0.6\%$ (SUN) on generalized ZSL.