[Reviews · NeurIPS 2020]

Review 1

Summary and Contributions: This paper proposes an attribute prototype network for zero-shot learning which improves local representations. The proposed network regresses and decorrelates attributes from intermediate features and outperforms other baselines.

Strengths: The proposed method is able to improve the locality representation of the zero-shot network. The effectiveness of the proposed method is verified through experiments and visualizations. The paper is clearly written and easy to follow.

Weaknesses: The proposed method is somewhat strongly correlated to Class Activation Map (CAM), which also uses attention-like locality to investigate and improve network representation. However, the relationship and differences between CAM and the proposed method is missing and CAM is only used for visualization. The idea of improving locality through attention/CAM is not new, and the proposed method lacks enough novelty. The authors should clarify their contribution against previous works.

Correctness: Yes

Clarity: Yes

Relation to Prior Work: Yes

Reproducibility: Yes

Additional Feedback: The proposed method is somewhat strongly correlated to Class Activation Map (CAM), which also uses attention-like locality to investigate and improve network representation. However, the relationship and differences between CAM and the proposed method is missing and CAM is only used for visualization. The idea of improving locality through attention/CAM is not new, and the proposed method lacks enough novelty. The authors should clarify their contribution against previous works. ----after rebuttal---- Thanks for the authors comprehensive rebuttal. It answered my question and addressed my concerns. I recommend acceptance.


Review 2

Summary and Contributions: A model for attribute-based zero-shot learning is proposed in this work. The main idea is to learn attribute prototypes in parallel to an object classifier used for ZSL. The prototypes help in learning better localized features (i.e. sensitive to the spatial location and extent of the attribute) which in turn is supposed to reduce the correlations among attributes and lead to better attribute predictions. Hence, by learning less correlated attributes the model generalization to unseen classes is improved. The model is evaluated on three public benchmarks for ZSL and GZSL showing consistent good performance. ======= Final Rating ====== After reading the other reviews and the rebuttal, I think the authors addressed my remarks adequately. I encourage the authors to include the new results, especially the binary attributes and attribute prediction performance in the final version. I think the submission is good based on the model design, performance, ablations and presentation. I recommend 7: Accept

Strengths: - The model requires attribute annotations at the class-level only. This enables the model to generalize to relatively large-scale data since this type of annotations is cheaper to obtain than fine-grained part annotations or image-level attributes and object bounding boxes. - In evaluation, the model shows good performance on three challenging datasets in both ZSL and GZSL setting. - An ablation study about the contributions of the different learning objectives is provided and shows that each of the introduced ideas is helpful in obtaining higher performance. - The analysis of the learned attribute prototypes in form of part localization and the qualitative experiments of the attention maps show that the attribute prototypes have descent performance in localizing the attributes in the input image. - Overall, the paper is well written and ideas were presented clearly.

Weaknesses: Novelty 1- The proposed model is mainly building on previous ideas: [8] for learning prototypes, [15] for decorrelation and sharing, [52] for localization compactness, and [7] for score calibration. This renders the technical novelty to be somewhat limited. Nonetheless, I find the employment of these ideas together for attribute localization and ZSL is quite interesting and seems to lead to consistent good performance. Model: 2- It seems that the model uses continuous attributes. This type of attributes is usually obtained averaging the image-level binary attributes for each class which is expensive to obtain. It would be interesting to see if the model work as well with binary attributes and what is the impact of these two representations on attribute localization. Evaluation: 3- Some state of the art methods that perform better than the current work were not reported in Table 2. For example, in GZSL setting [A] and [B] show better performance in SUN and [C] shows better performance in AwA2. 4- The model is evaluated on ZSL, GZSL and part localization; however, attribute prediction is a relevant task where the model should show improved performance as well due to its ability to localize relevant features. There were no evaluation of this aspect, it would be interesting to see if APN leads to better attribute predictions at the image level (e.g. in CUB) for the seen and unseen classes and how it compares to a standard CNN model trained for attribute prediction without prototypes. [A] Generative Dual Adversarial Network for Generalized Zero-shot Learning. CVPR 2019. [B] f-VAEGAN-D2: A Feature Generating Framework for Any-Shot Learning. CVPR 2019. [C] Rethinking Zero-Shot Learning: A Conditional Visual Classification Perspective. ICCV 2019.

Correctness: The claim of outperforming SOTA on the three benchmarks and on both ZSL and GZSL setting seems partially correct. Some methods with better performance on some of the datasets were not reported (see Weaknesses) The method looks correct.

Clarity: The paper is well written.

Relation to Prior Work: The related work discussion seems adequate.

Reproducibility: Yes

Additional Feedback: Please address the points raised in Weaknesses.


Review 3

Summary and Contributions: The paper proposes a network of localizing attributes in images for zero-shot learning. The zero-shot learning model with the supervision of localized attributes regression, can reach state-of-the-art accuracy on benchmark datasets. Qualitative results are illustrated to show the model is able to localize object parts without location annotations during training.

Strengths: 1. The paper proposes to learn prototypical representations for each attributes to localize the visual representations of attributes. Furthermore, for attributes encoding the semantic part information (like in CUB), the proposed method can accurately localize the semantic parts without explicit supervision. 2. The proposed ZSL method is evaluated extensively on benchmark datasets and compared to state-of-the-art methods. It outperforms SOTA methods with considerable margins. 3. The authors illustrate extensive qualitative results to verify the ability of learning accurate part locations. Quantitative results are also provided to show its localization accuracy.

Weaknesses: 1. The novelty of the paper is limited. Learning attribute localization with divergence and concentration losses (L_{AD} and L_{CPT}) for ZSL has been explored in other works like [58]. Similar ideas have also been proposed in previous works like [R1], where the authors also learns part localization and attribute prototypes without explicit supervision and applies to zero-shot learning. I would encourage authors elaborate the differences between these works and clarify the specific novelties and contributions proposed in the paper. 2. The claimed effect of ProtoMod needs more evaluations. The authors provide some qualitative results in Figure 3 and Fig Q.3 & Q.4 (from supp.) to show the proposed ProtoMod can localize the attributes like 'black back' and 'yellow breast'. What concerns me is whether the model is indeed learning the specific attributes (black back) or only the partial semantic like 'black' or 'back'. The part localization experiments (Fig 2 and Table 3) may partially show the model can learn the part information ('back', 'belly') from attributes but does not show if the model can distinguish between 'black back' or 'yellow back'. Neither the attribute similarity maps from Fig 3 and Fig Q.3 & Q.4. I think some qualitative results of the same images with locations of attributes only differ partially (e.g. 'black back' vs. 'yellow back' vs. 'black breast') can further verify whether the ProtoMod is learning the specific attributes or just the color or parts. --------- Ref: R1: Zhu, Pengkai, Hanxiao Wang, and Venkatesh Saligrama. "Learning classifiers for target domain with limited or no labels." ICML 2019. --------- Updates: the authors address my concerns in the rebuttal. I raise my score to accept.

Correctness: The proposed method is overall correct. The claims need more evaluations as I described above.

Clarity: The paper is written clearly and easy to follow.

Relation to Prior Work: The paper needs clear discussions on how the proposed method differs from previous works using the same strategy to learn attribute locations and prototypes with weak supervision for zero-shot learning. Details above.

Reproducibility: Yes

Additional Feedback:

[Author Response · NeurIPS 2020]



**Figure 1.** Localizing partially different attributes, e.g. "white head" v.s. "black head".

| Method | CUB | AWA2 | SUN |
|---|---|---|---|
| [A] | 49.5 | 43.5 | 53.4 |
| [B] | 68.9 | 65.2 | 43.1 |
| [C] | 47.5 | 66.7 | - |
| APN+[A] (Ours) | 67.3 | 46.5 | **56.7** |
| APN+[B] (Ours) | **70.0** | 65.6 | 43.7 |
| APN+[C] (Ours) | 67.7 | **71.0** | - |

**Table 1.** Results of applying our learned APN features to [A], [B] and [C] in GZSL (harmonic mean reported). We did not reproduce results of [C] on SUN due to time limit (will include in final paper).

**R1, R2, R3: novelty.** We thank the reviewers for mentioning a few pioneering works [8, 58, D] that learn to localize object parts. Here we compare our work with these papers and clarify our novelty. [8] learns multiple prototypes for each object class with image labels to improve the model interpretability but is not tailed for ZSL. However, we aim to improve the image representation for ZSL by learning attribute prototypes with class-level attributes. [58] proposes to improve the image features by learning channel-wise part attention. Similarly, [D] uses the channel grouping model [52] to learn part-based representations and part prototypes. In contrast, we treats each channel equally, and we use spatial features associated with input image patches to learn prototypes for attributes (see Fig.1 in the main paper). We argue that these methods are limited when localizing object parts and visual attributes. Specifically, they learn latent attention/prototypes during training, whose meaning is posteriorly inducted by observation, which is not deterministic. Besides, they can only localize a small number of object parts, i.e., [58] for 2 parts, [D] for 4 parts, and are not able to localize attributes, which play an important role for ZSL. Very different from these publications, our work is innovative in two aspects which overcome the above mentioned limitations. 1) We improve the attribute localization and image features by learning prototypes to regress attributes, where each prototype corresponds to a specific attribute. Therefore, our model can localize all the attributes and parts, which is essential in building trustworthy ZSL model to associate attributes with correct image regions. Our claim can be supported by the results in Fig. 1 and other results (Fig. 2, 3, Tab. 3) in main paper. 2) We decorrelate prototype learning and enforce the localization compactness. In R2's words, "the employment of these ideas together for attribute localization and ZSL is quite interesting and seems to lead to consistent good performance" (see Tab. 1 for improvement over SOTA). The discussion will be added in the final paper.

**R1: differences with CAM.** CAM is a post-hoc method to investigate the model attention by computing the channel-wise weighted sum of last layer CNN feature maps, which didn't improve the image feature. In contrast, our APN aims to improve the image representation for zero-shot learning by learning prototypes that predict attributes from intermediate features. Besides, while the original CAM generates one global attention map for predicting an object class in a given image, our APN can generate multiple attention maps that localize different attributes in an image. Our APN can obtain attention maps that better localize visual attributes compared to CAM (See Fig. 3 in the main paper).

**R2: impact of binary and continuous attributes.** We follow [21] to generate class binary attributes by thresholding the continuous attributes. Our method works equally well for both kinds of attributes in terms of ZSL accuracy (continuous $73.3\%$ vs binary $73.1\%$ on CUB) and part localization accuracy (continuous $52.8\%$ vs binary $52.1\%$ on CUB).

**R2: evaluate attribute prediction** As suggested by R2, we build an attribute prediction baseline by training a standard CNN to predict binary attributes without prototypes. Our APN (also trained with binary attributes) can classify binary attributes with a threshold of $0.5$. The evaluation is conducted on the image-level attributes of test images on CUB dataset. Our APN achieves attribute prediction accuracies of $87\%$ on unseen classes and $90\%$ on seen classes, which significantly outperforms the baseline model without prototype that obtains $82\%$ (unseen) and $84\%$ (seen).

**R3: ProtoMod needs more evaluation.** For each attribute, we learn a prototype on intermediate CNN features to regress the desired attribute. So it is expected to learn specific attributes rather than just the color or parts, which is empirically confirmed by the attribute prediction accuracy which is higher than the attribute prediction model (See previous response to R2), and the qualitative results in Figure. 1, where our APN is able to precisely localize "white head", "black head" and "black belly" in the first image and "yellow belly", "white leg" and "white belly" in the second.

**R2: comparing with [A], [B] and [C].** While [A], [B] and [C] propose strong ZSL classifiers with fixed ImageNet-pretrained (or finetuned) features, our APN aims to improve the image features for ZSL. In Table 1, we show new SOTA GZSL results when applying the image features extracted from our learned APN to [A], [B] and [C]. For example, on SUN, our APN+[A] achieves $56.7\%$ vs $53.4\%$ of [A]. On CUB, our APN+[B] obtains $70.0\%$ vs $68.9\%$ of [B]. On AWA, our APN+[C] reaches $71.0\%$ vs $66.7\%$ of [C]. We will include these papers and results in the final paper.

[A] Huang et al., Generative Dual Adversarial Network for Generalized Zero-shot Learning, In CVPR 2019

[B] Xian et al., f-VAEGAN-D2: A Feature Generating Framework for Any-Shot Learning, In CVPR 2019

[C] Li et al., Rethinking Zero-Shot Learning: A Conditional Visual Classification Perspective, In ICCV 2019

[D] Zhu et al., Learning classifiers for target domain with limited or no labels, In ICML 2019


[Meta-Review · NeurIPS 2020]

The authors satisfactorily addressed the concerns of the reviewers regarding novelty, comparisons to related work, and visualizations of mixed-attribute localization. As such, it is felt that this paper would be a good contribution to the ZSL field. Please incorporate the exposition, clarifications, and additional experiments from the rebuttal into the main text.